# Development of a Probiotic Beverage Using Breadfruit Flour as a Substrate

**DOI:** 10.3390/foods8060214

**Published:** 2019-06-17

**Authors:** Yifeng Gao, Nazimah Hamid, Noemi Gutierrez-Maddox, Kevin Kantono, Eileen Kitundu

**Affiliations:** Department of Food Science, Auckland University of Technology, Private Bag 92006, Auckland 1142, New Zealand; stanleygyf@gmail.com (Y.G.); ngutierr@aut.ac.nz (N.G.-M.); kkantono@aut.ac.nz (K.K.); ekitundu@aut.ac.nz (E.K.)

**Keywords:** breadfruit, probiotic foods, lactic acid bacteria, non-dairy functional beverage

## Abstract

A fermented beverage was developed using breadfruit flour as a substrate by optimising sucrose, inoculum concentrations, and fermentation temperature in the formulation by utilising the D-optimal mixture design. The optimisation was carried out based on CFU counts, pH, titratable acidity, lactic acid, and sugar concentration of the different fermented breadfruit substrate formulations. Results showed that the optimised values based on the contour plots generated were: 7% breadfruit flour, 1% inoculum, and 15% sugar after fermentation at 30 °C for 48 h. Sensory projective mapping results showed that the fermented breadfruit substrate beverage was characterised by a pale-yellow appearance, fruity flavour, and sweet and sour taste. The hedonic test was not significantly different (*p* > 0.05) for almost all formulations except for formulation 4 (5% sugar, 3% inoculum, 7% breadfruit flour at 30 °C), which was described as bitter and had the lowest acceptance rating. This study successfully demonstrated the development of a novel fermented breadfruit-based beverage with acceptable sensory characteristics and cell viability using a mixture strain of *L. acidophilus* and *L. plantarum* DPC 206.

## 1. Introduction

The global market of probiotic foods in the 21st century has been estimated to be worth over 40 billion USD, with consumption forecasted to exceed 12 million tons by 2024, and an estimated growth potential of 11.7% per year [1,2,3]. The most common functional foods manufactured are probiotic foods and beverages. It has been recorded that probiotics can enhance immune system function, nourish beneficial intestinal flora, stimulate their development, reinforce their action, and assist in the absorption of vitamins and minerals [4]. For health benefits, the minimum dose of 10^7^ colony forming per unit (CFU)/g or mL probiotic bacteria in food is recommended [5,6]. Strains of lactic acid bacteria (LAB) belonging to the genera *Lactobacillus* and *Bifidobacterium* are commonly used as probiotics.

Commercial functional foods in the food market are mainly dairy-based products, although consumers are increasingly requiring new functional products that are non-dairy [7]. With the increase in lactose intolerance and allergies, attempts had been made to develop fruit-based foods as an alternative to traditional dairy functional foods [7]. 

Breadfruit (*Artocarpus altilis*) is a nutritious fruit cultivated by Pacific Islanders for over 3000 years and is abundant in Polynesia, Jamaica, and the Caribbean Islands [8]. Breadfruit belongs to the *Moraceae* family and consists of more than 50 species. Amusa et al. [9] reported that breadfruit can be propagated through stem-cuttings and the first fruiting average period of the crop is from four to six years. Every year, a single breadfruit tree produces 150–200 kg of fresh fruit. The fruit is usually ovoid or oblong [10]. It is now recognised as a staple food and can be consumed cooked, roasted, fried, boiled, dried, or pickled [11]. 

Breadfruit is high in carbohydrate and contains protein, 2.2–5.9% on a dry weight basis. It has been known to be a good source of amino acids, specifically histidine and lysine, which are crucial for infant growth. Depending on the cultivar, breadfruit has been reported to be a rich source of vitamins and minerals such as copper, magnesium, phosphorous, potassium, calcium, iron, and manganese [12,13,14]. Breadfruit also contains various bioactive compounds such as phytate, oxalate, and tannin [15], ascorbic acid [16], and high contents of carotenoids [17]. 

Although breadfruit is nutritious, it rapidly undergoes physiological deterioration after harvesting. As a way of minimising post-harvest losses and increasing the utilisation of breadfruit, the fruit can be processed into flour, which is more shelf stable. The suitability of the flour for use in food depends on its functional properties. Interestingly, breadfruit flour had a higher water content, oil absorption capacity, foaming capacity, and emulsion activity compared to wheat flour [18] and yam flour [19].

Considering the rich nutritional content and health benefits of breadfruit as a non-dairy fermented media, this study aimed to investigate lactic acid fermentation using breadfruit flour as a substrate, and monitor the changes in terms of physicochemical (pH, organic acids, sugar concentration), and sensory properties of selected fermented beverages.

## 2. Materials and Methods

### 2.1. Breadfruit Flour 

Breadfruit flour was sourced from a local company (Maiden South Pacific Company) located in Auckland, New Zealand. The breadfruit flour was imported from Natural Foods International Limited, Samoa. Mature breadfruit were harvested, peeled, cored and sliced before being air dried, and then hand milled into powder form (G. Percival, personal communication, 25 January 2019).

### 2.2. Microorganisms

Three lactic acid bacteria were used in this research, namely *Lactobacillus plantarum* DPC 206 (Bioactive Research, Auckland, New Zealand), *Lactobacillus acidophilus* “de Winkel” (De Winkel yoghurt, Fonterra Cooperative Group, Auckland, New Zealand) and *Lactobacillus casei* Shirota (Yakult, Tokyo, Japan). All lactic acid bacteria were grown in MRS broth (Merck, Darmstadt, Germany). The bacteria were selected as they were common starter bacteria in existing commercial fermented products.

### 2.3. Preparation of Lactobacilli-Fermented Breadfruit Beverages

#### 2.3.1. Preliminary Studies on the Selection of *Lactobacillus* Strains 

Each strain from the stock cultures was activated prior to preparation of the breadfruit-based beverage. This was done by adding 1% *v*/*v* inoculum into breadfruit supernatant and incubation at 37 °C for 24 h. The steps in making the fermented breadfruit beverages are summarised in Figure 1. During fermentation, aliquot of samples (50 mL) were taken at 0, 12, 24, 48, and 72 h intervals for chemical and microbiological analyses. A selection of lactic acid strains to be used in probiotic beverage production was performed based on the viable cell number and preliminary sensory quality evaluation.

#### 2.3.2. Viable Cell Count Determination

Determination of the number of viable cells was carried out using the plate count technique. Suspensions of fermented beverage were decimally diluted in sterile peptone water up to 10^−5^ dilution. Aliquots of the diluted fermented beverage (0.1 m) L were inoculated on De Man, Rogosa and Sharpe (MRS) agar plates using the spread plate method (Merck, Darmstadt, Germany). The plates were incubated at 37 °C for 72 h in the CO_2_ incubator. Plates were counted manually and recorded as colony-forming units (CFU) per mL of culture. Viable cell count was obtained using triplicate plates for each beverage sampled periodically at 0, 12, 24, 48, and 72 h. 

### 2.4. Preliminary Sensory Evaluation of Samples

Preliminary sensory evaluation of the fermented breadfruit beverage was carried out in a sensory testing facility. Seven beverage samples with the maximum number of viable cells were chosen. Samples were served in plastic portion cups labelled with three-digit random numbers. Twelve panellists described differences between the samples of fermented beverages in terms of smell, colour and taste. 

### 2.5. Determination of pH

The pH of beverages was determined throughout the fermentation using a digital pH meter (Eutech pH 700 meter, Thermo Fisher Scientific Inc., Waltham, MA, United States) with a glass electrode (Electrode ECFC7252101B, Thermo Fisher Scientific Inc, Waltham, MA, United States). Before measurement, the pH meter was calibrated with buffers (Thermo Fisher Scientific Inc, New Zealand) at pH 4.0 and 7.0. pH determination was performed in triplicates for each fermented beverage sample (20 mL) at 0, 12, 24, 48, and 72 h. 

### 2.6. Experimental Design for the Optimisation of the Fermentation Process for the Production of Probiotic Beverages

The D-optimal design (value 0.95) was applied to investigate the influence of breadfruit, sugar, temperature and inoculum concentrations, using the Unscrambler X v10.1 (CAMO ASA, Oslo, Norway) software. This design minimises the covariance of the parameter estimates for a specified model [20]. In this experimental design, breadfruit (2% to 7%), sugar (5% to 15%), temperature (30 to 37 °C, and inoculum (1% to 3%) were used as experimental variables. D-optimal design was utilised with the constraints for the following: Sugar (*X*_1_) + Inoculum (*X*_2_) + Breadfruit (*X*_3_). Unscrambler designed 19 runs with two replicates (Table 1).
*Y* = λ_1_*X*_1_ + λ_2_*X*_2_ + λ_3_*X*_3_ (linear),*Y* = λ_1_*X*_1_ + λ_2_*X*_2_ + λ_3_*X*_3_ + λ_1_X_1_λ_2_*X*_2_ + λ_1_*X*_1_λ_3_*X*_3_ + λ_2_*X*_2_λ_3_*X*_3_ (quadratic),*Y* = λ_1_*X*_1_ + λ_2_*X*_2_ + λ_3_*X*_3_ + λ_1_X_1_λ_2_*X*_2_ + λ_1_*X*_1_λ_3_*X*_3_ + λ_2_*X*_2_λ_3_*X*_3_ + λ_1_*X*_1_λ_2_*X*_2_λ_3_*X*_3_ (special cubic),(1)
where *Y* represents the responses of the experiment (CFU, pH, Titratable acidity, Lactic Acid, and Sugar), λ is the constant coefficients, and *X* is the proportions of the components.

### 2.7. Determination of Titratable Acidity

Titratable Acidity (TA) of fermented beverages was carried out using the AOAC 937.05 method [21]. 0.1 N NaOH solution was used as titration solution. The percentage titratable acidity, as lactic acid was determined using the following equation:(2)$$\%\;\mathrm{Titratable}\;\mathrm{Acidity},\;\mathrm{as}\;\mathrm{lactic}\;\mathrm{acid}=\frac{N\times V\times 90.08}{W\times 10},$$
where: *N* = normality of titrant, 0.1 N NaOH; *V* = volume of titrant (mL); *W* = mass of breadfruit substrate beverage (g). 

### 2.8. Determination of Sugar Concentration

High-Performance Liquid Chromatography (HPLC, Agilent Technologies, Inc., Santa Clara, CA, USA) was used to analyse the sugars in the 48-h-fermented beverage samples based on the AOAC 980.13 method [22]. This method was capable of detecting fructose, glucose, lactose, maltose and sucrose sugars. However, sucrose was the only sugar detected in the samples, and thus the only sugar reported in this study. In each run, sugars were quantified by a R401 refractive index detector, and separated on a Shodex Asahipak (250 × 4.6 mm) column. The mobile phase used was a 80% acetonitrile solution that was passed through the column at a flow rate of 1.5 mL/min. Prior to injection, samples were centrifuged for 10 min at 2000 rpm and filtered through a 0.45-µm Swinney syringe filter.

### 2.9. Determination of Lactic Acid Concentration

The methyl chloroformate (MCF) method was used in this study to derivatise metabolites [23]. This method involved a fast alkylation reaction, where amino acid and non-amino organic acids are quickly reacted with MCF to form esters and carbamates [23]. Derivatised samples were analysed using the gas chromatography–mass spectrometry (GC-MS) method (Model 5977B, Agilent Technologies Inc.), equipped with a column (Model122-5532G, length 30 m, diameter 0.250 mm, film 0.25 µm, Agilent Technologies Inc.). After MCF derivatisation, GC-MS analysis was carried out using a temperature program that started at 30 °C and held for 4 min, followed by a 10 °C/min increase to 250 °C, and maintained at 250 °C for 3 min. The mobile phase used was helium at a flow rate of 54.4 mL/min. 

### 2.10. Sensory Evaluation

#### 2.10.1. Projective Mapping

The projective mapping of samples in this study followed the protocol described by Balbas et al. [24]. In addition to differentiating products in terms of their similarities and differences, products were also described in terms of their sensory attributes. Product profiles from projective mapping have been reported to show a high degree of similarity to results obtained by descriptive analysis [25]. In this study, projective mapping was carried out by a semi-trained panel that comprised of 17 panellists (aged from 20 to 29, with equal numbers of males and females). Six different beverage samples fermented for 48 h were selected. Panellists were served samples (20 mL each) in portion cups labelled with random three-digit numbers and served in a random order at room temperature. Panellists were also asked to describe attributes that differentiated the beverage samples.

#### 2.10.2. Consumer Testing

Consumer testing was carried out on 50 consumers. Consumers rated their response to the beverages in terms of overall liking, and liking of appearance, odour, flavour, texture and aftertaste using a 15-cm unstructured line scale, anchored “extremely dislike” on the left and “extremely like” on the right. Panellists were served samples (20 mL each) in portion cups labelled with three-digit random numbers and served in a randomised order at room temperature. Data were collected using the FIZZ Acquisition system v 2.46c (Biosystèmes, Couternon, France). 

### 2.11. Statistical Analysis. 

Two-way analysis of variance was used to explore the main and interaction effects of fermentation time and bacterial species mix based on CFU counts and pH. Product optimisation was carried out using the d-optimal design. In addition, contour plots were generated to explore the relationship between sugar, breadfruit flour, and inoculum concentration. All product optimisation statistical analysis was carried out using the Unscrambler X v10.1 software (Camo Analytics AS, Oslo, Norway). 

Analysis of projective mapping results was performed according to Balbas et al. [24] using XLSTAT version 2018.1 (Addinsoft Inc, Brooklyn, NY, USA). Multiple Factor Analysis (MFA), Generalised Procrustes Analysis (GPA), and Principal Component Analysis (PCA) were carried out to 1) determine panel agreement, 2) extract product coordinates, and 3) visualise product coordinates with correlative sensory attribute (at a minimum frequency of five times). 

## 3. Results and Discussion

### 3.1. Growth of Lactic Acid Bacteria 

Three probiotic strains (*L. acidophilus*, *L. casei* and *L. plantarum* DPC 206) were selected for fermentation of a probiotic drink using a water extract of breadfruit flour fermented at 37 °C from 0 to 72 h. The viability of cells during fermentation are presented in Table 2. After fermentation from 48 to 72 h, the maximum number of *L. acidophilus*, *L. casei* and *L. plantarum* DPC 206 were between 7.931 and 8.029 log_10_ CFU/mL respectively with no significant difference (*p* > 0.05). The most rapid growth occurred with *L. acidophilus*, which started off with the lowest viable cell number and reached a maximum of 8.029 log_10_ CFU/mL after 72 h fermentation. *L. casei* and *L. plantarum* DPC 206 showed similar maximum cell viability in the fermented beverage. *L. plantarum* DPC 206 started with a relatively low viable cell number and reached a maximum after two days fermentation, while numbers of *L. casei* reached a maximum after 72 h fermentation. 

In the four groups of mixed strains fermentation (*L. acidophilus* + *L. casei*, *L. acidophilus* + *L. plantarum* DPC 206, *L. casei* + *L. plantarum* DPC 206, and *L. acidophilus* + *L. casei* + *L. plantarum* DPC 206), the maximum number of the mixed *Lactobacilli* were between 7.962 and 8.238 log_10_ CFU/mL. The beverages fermented with a mixture of three *Lactobacilli* presented the highest cell counts (8.238 log_10_ CFU/mL) compared to those fermented with a mixture of two *Lactobacilli*. *L. acidophilus* with *L. casei* and *L. acidophilus* with *L. plantarum* DPC 206, but showed similar characteristics in bacteria growth. During the fermentation, both groups reached maximum viable counts, with *L. acidophilus* and *L. casei* showing a higher growth at 8.014 Log_10_ CFU/mL than a mixture of *L. acidophilus* and *L. plantarum* DPC 206 (7.962 Log_10_ CFU/mL), but with no significant difference (*p* > 0.05). After 24 h fermentation, the viability of both groups showed a moderate decrease between 7.435 and 7.701 Log_10_ CFU/mL. The *L. casei* and *L. plantarum* DPC 206 group as well as the mixture strains group of three *Lactobacilli* at 48 h fermentation, showed similar bacteria growth and achieved high number of viable cell counts of 8.22 Log_10_ CFU/mL and 8.238 Log_10_ CFU/mL, respectively.

For all seven groups of bacteria culture or their mixtures, the cell concentration of samples showed no significant difference (*p* > 0.05) when fermented for 72 h. A significantly (*p* < 0.05) low cell concentration of *L. acidophilus* was found at 24 h, compared to other groups. Significantly (*p* < 0.05) rapid growth occurred for most strains for *L. acidophilus* and *L. plantarum* DPC 206 from 0 to 12 h. Cell viability increased in the early stage of fermentation and contained enough probiotics (7 log_10_ CFU/mL). This result was in agreement with previous studies [26,27]. Mousavi et al. [28] reported that, once lactic acid bacteria, such as *L. acidophilus* and *L. plantarum*, successfully grew under new conditions, they enter the exponential growing phase. For all seven groups, no significant difference (*p* > 0.05) was observed between 12 and 24 h. When probiotic cell counts increased to a maximum, the viability of probiotic bacteria decreased slightly due to the production of inhibitory substances, such as lactic acid [3]. Usually, the growth capacity of *L. acidophilus* and *L. plantarum* mainly depend on the nutrient content in the medium [29]. Probiotic species and fermentation time were however significantly influenced in terms of cell concentration (F value 5.82 **, 96.7 **, *p* < 0.01 in Table 2, respectively). 

For all seven groups of individual probiotic bacteria and their mixtures, the total number of viable cells were over 7 Log_10_ CFU/mL in the final product. Thus, the results demonstrated that the selected *Lactobacilli* were able to grow in breadfruit substrate beverages successfully. The results also showed that the mixed culture of lactic acid bacteria grew faster than single lactic acid bacteria strains in the breadfruit substrate beverages. Mixed cultures have been reported to contain more than the recommended probiotic cell level (7Log_10_ CFU/mL) after fermentation [5]. In general, mixed cultures are involved in the interaction mechanism that may stimulate or inhibit bacteria growth [30]. Mixed strains presented fast growth that could be due to the interaction mechanism that can produce different metabolites. For example, *L. acidophilus* is a homofermentative bacteria that produces lactic acid by glycolysis ([31]. *L. plantarum* is a heterofermentative bacteria that produces lactic acid and other end-products [31]. During fermentation, these metabolites can stimulate the growth of mixed cultures. In conclusion, the three *Lactobacilli* strains and mixed strains used in this study exhibited good adaptation to the breadfruit substrate, and the viability of cells in the fermented beverages yielded a satisfactory probiotic value.

### 3.2. Acidification in Fermented Breadfruit Beverage

The change in pH values during fermentation is presented in Table 3. For all the beverages containing three probiotic strains and mixture strains, fermentation started at a similar pH and dropped between 4.62 and 3.49 at the end of 72 h fermentation. This decrease in pH can be due to the lactic acid bacteria producing organic acids, which is mainly lactic acid. *L. acidophilus* pH was significantly (*p* < 0.05) higher than other strains at all fermentation times of 12, 24, 48 and 72 h. Interestingly, although a higher cell growth was found with *L. acidophilus* at all fermentation times, a lower acidification was produced. This can be explained due to differences in metabolism and production of organic acids in the different strains [32]. The pH of mixture strains containing *L. plantarum* DPC 206, except for a mixture of *L. acidophilus* and *L. plantarum* DPC 206 strains, were significantly (*p* < 0.05) the lowest at 48 and 72 h. The type of probiotic bacteria, fermentation time, and their interaction significantly influenced pH (F value 92.1 ***, 549.8 ***, and 5.5 ***, *p* < 0.0001 in Table 2). 

### 3.3. Preliminary Sensory Evaluation

Preliminary sensory testing was carried out to screen a suitable starter culture for the development of the probiotic beverage. The fermented beverages fermented by *L. casei*, and a mix of *L. casei* and other strains were found to have an undesirable smell that panellists found unacceptable. The negative perception of *L. casei* fermented beverage might be due to probiotic off-flavour and higher lactic acid content that can decrease acceptability. *L. casei* fermented with litchi juice has also been reported to result in unfavourable flavour amongst panellists [33].

Both *L. plantarum* DPC 206 and a mix of *L. acidophilus* and *L. plantarum* DPC 206 fermented beverages were reported to have a fruity character and pleasant flavour that panellists found favourable. However, since the *L. acidophilus* and *L. plantarum* DPC 206 mix strains in the beverage was fast growing, and only required 48 h of fermentation to reach a maximum viable cell count, this inoculum was used for further optimisation experiments for production of a fermented breadfruit substrate beverage. In fermentation procedures, it is desirable to have short fermentation periods as this can enhance output and prevent microbial contamination [34]. The co-cultured organisms grew the quickest under these conditions to gain ascendance and predominated because organisms must compete for nutrients or produce metabolites that stimulate each other’s growth [35]. In addition, *L. plantarum adapt* DPC 206 has been reported to do well in various environments because of its metabolic flexibility [36].

### 3.4. Optimisation of Fermented Breadfruit Beverage Using a Mixture Design Experiment

Cell viability results in this study, decreased linearly with temperature increase, with no significant (*p* > 0.05) effect. Hence optimisation of the fermented beverage was further carried out using CFU, pH, TA, LA and sugar concentration instead. Our results are in agreement with previous researches [37,38] for vegetable juice and cashew apple juice [39]. In their study, mixed probiotic strains were observed at moderate fermentation temperature, with maximum cell viability at 30 °C. Temperatures higher than 30 °C can cause viability losses.

The D-optimal mixture experimental design is often applied in food fermentation as it is an effective tool for optimisation [40]. This design was employed in this research using a mixed culture grown at 30 °C after 48 h fermentation. Seven percent breadfruit flour was used because proportions higher than that resulted in a more viscous product that cannot be fermented. Sugar was added at 15% to the fermented beverage to give a balanced sweet and sour taste, as recommended by the focus group that carried out preliminary sensory testing who also found that the mixture strains of *L. acidophilus* and *L. plantarum* DPC 206 resulted in acceptable fermented beverage sensory attributes. 

Experiments runs were generated using the Unscrambler X v10.1 (CAMO ASA) software. The fitted models obtained for each response were fitted to a model based on SS and *R*^2^. Table 4 presents the equations and adjusted coefficients of determination of models. Results showed that the five response variables measured (CFU, TA, pH, LA, and sugar concentration) belonged to either quadratic, quartic and special cubic models (Table 5). The polynomial models that explained the relationship between response and the variables are presented in Table 4. 

#### Optimisation of Five Response Variables (CFU, pH, TA, LA and Sugar Concentration) in Breadfruit Substrate Beverage Fermented for 48 h

In Figure 2, the mixture contour plots presented a two-dimension view wherein all points located in the same shade regions are related to the cubic model. The effect of sugar, inoculum and breadfruit flour concentration and their interactions were investigated to understand the changes in growth of *L. acidophilus* and *L. plantarum* DPC 206 using a cubic model. Each side of the triangle represented maximum values of fermentation parameters and the opposite side represented the minimum value. The regression model equations for CFU, pH, titratable acidity, lactic acid and sucrose concentration are presented in Table 4.

As seen in Figure 2a, the area with the highest CFU was located on the right-hand side of the triangle plot. The maximum value was located near the top region of this line. Decreasing breadfruit and inoculum contributed to a significant (*p* < 0.05) increase in CFU of fermented beverage. Sugar content was also found to significantly (*p* < 0.05) increase CFU in the fermented breadfruit beverage. Angelov et al. [26] similarly showed that increasing sugar concentration enhanced cell growth. In their study, a significantly higher cell count was observed with 2% sucrose at 2.81 log orders compared to 1.5% and 1% sucrose for the newly developed oat-based probiotic drink.

As seen in Figure 2b, the area with the highest pH was located on the right-hand side of the triangle plot. The maximum pH region was located midway. pH reached a maximum at around 5.5% breadfruit and 13% sugar concentration. According to Table 5, only breadfruit proportion and sugar concentration significantly (*p* < 0.05) influenced pH values. The optimum value of pH in the contour plot was found at 3.88. Kailasapathy and Chin [41] pointed out that pH values between 3.5–4.5 increased the stability of probiotics in the gastrointestinal tract, which enhances survival of probiotic strains consumed. Although lower pH resulted in probiotic bacteria loss, *L. acidophilus* and *L. plantarum* were able to tolerate the lower pH because a proton gradient existed in the cell in order to counteract the large amount of lactate in the food medium [42]. 

As seen in Figure 2c, the area with the highest TA was located at the middle range of the triangle plot. The increase in TA was only significantly (*p* < 0.05) affected by sugar and inoculum interaction (Table 5). The maximum value of TA in the contour plot was 0.2%. These TA values were similar to those found in fermented soy-based products (0.08–0.19%) [43]. The differences in TA value may be due to different nutrient content as well as the different fermentation parameters used for the different probiotic strains [30]. 

As seen in Figure 2d, the area with the highest LA was located midway on the right-hand side of the triangle plot. LA was at a maximum with around 5.5% of breadfruit and 13% sugar concentration. The highest value of LA was 0.89 g/mL. Results showed that 5% of sugar was enough for lactic acid bacteria to grow and accumulate lactic acid. Significance was only observed in the quartic interaction (sugar × breadfruit × (sugar – breadfruit)^2^) as seen in Table 5. Sugar and breadfruit had the most significant effect on LA at a mid-range content (also known as a turning point in polynomial equations). On contrary, the lowest LA points were observed in the lowest and highest content of sugar and breadfruit, respectively. 

As seen in Figure 2e, the area with the highest sucrose concentration was located midway on the left-hand side of the triangle plot. The maximum sucrose concentration region was located in the middle range of sugar and inoculum. According to Table 5, it was the interaction between sugar and inoculum that significantly influenced sucrose concentration. Maximum sucrose concentration was found in formulations with mid-level concentrations of inoculum (1.5%) indicating that higher counts of starter culture resulted in low viable growth rate [26]. 

The data shown in Table 5 compared the experimental value with the D-optimal prediction value. Predicted values were calculated for the optimised design based on CFU, pH, TA, LA, and sucrose concentration. In this study, CFU was set as the most important variable while TA, LA, and sucrose concentration results were set as second priority variables. The optimum experimental values of CFU, pH, and LA were slightly lower than the predicted values, except for TA and sugar concentration. Overall, the optimum fermentation parameters for the fermented breadfruit beverage were found to be 7% breadfruit, 15% sugar and 1% sugar on 48 h fermentation at 30 °C, based on the optimised results using the D optimal design. 

## 4. Sensory Quality Evaluation

Six of the 19 formulations (Formulations 1, 2, 3, 4, 6 and 18) in the experimental design that had high viable counts were subjected to sensory testing. The selected formulations were those fermented for 48 h. 

### 4.1. Projective Mapping

Figure 3 shows the results of descriptive sensory attributes of beverages from formulations 1, 2, 3, 4, 6 and 18 obtained from sensory projective mapping based on appearance, aroma, taste and flavour attributes. As seen in Figure 3, a total of 58.64% of the variation between samples was explained. The first axis explained 31.35% of the total variation, and the second axis up to 27.29% variance. The first component (F1) separated bitter from sour, honey, fruity and sweet. As for the second axis, appearance characteristics of opaque were separated from pale yellow. 

According to Figure 3, Formulations 1, 3 and 6 were characterised primarily by mint, sour, creamy appearance, honey, fruity flavour and sweet. Formulation 4 was mainly separated by the appearance—pale yellow and bitter. Costa et al. [44] reported that juices with added sugar that tasted sweet helped reduce the perception of sour. Formulation 4 presented a bitter taste that may have been caused by some metabolites produced during fermentation. This could be due to the long fermentation time (48 h). Bitter peptides (peptides aS1-CN) in the beverages have been reported to contribute to bitterness [32]. Lactic acid bacteria growth can lead to consumption or formation of compounds that may change flavour or aroma [1]. 

### 4.2. Sensory Acceptance 

Figure 4 summarises the results of the hedonic test. There were no significant differences in acceptance (*p* > 0.05) among the different formulations when evaluated for appearance and odour. This indicated that different fermentation conditions and sugar addition did not affect the appearance and odour of breadfruit beverage. Sensorially, Formulation 4 was consistently significantly lower (*p* < 0.05) in acceptability than the other formulations based on liking of appearance, flavour and aftertaste, as well as overall liking. Formulation 4 happened to contain low sugar and higher concentration of cultures, which may explain why it was the least acceptable. Other studies on probiotic cashew apple juice have reported that increasing sucrose led to increased overall taste acceptance [45]. In addition, Formulation 4 (7% breadfruit, 3% inoculum and 5% sugar) was also described as being bitter from the projective mapping results. According to Cruz et al. [46], metabolites from lactic acid bacteria can negatively contribute to the aroma, off-flavour and taste of a probiotic product. Hence, the amount of sucrose added is critical to the acceptability of the fermented breadfruit beverage developed in this study.

## 5. Conclusions

This study demonstrated the use of breadfruit flour as a novel substrate for fermentation to produce a non-dairy probiotic beverage. The beverage formulated was found to have acceptable sensory characteristic and good cell viability using a mixture strain of *L. acidophilus* and *L. plantarum* DPC 206. The optimisation for production of the fermented beverage in terms of CFU, pH, titratable acidity, lactic acid and sucrose concentration was successfully achieved using the D-optimal mixture design approach. Sensory characterisation revealed that the beverage had favourable sensory characteristics with good consumer acceptability. The market demand for non-dairy fermented beverages is mainly driven by the increasing number of health-conscious consumers demanding for food products with added value and improved functionality. With increased awareness of the health benefits of consuming beverages that are naturally fermented, the market for these products is expected to increase in health-oriented Western countries. Future research should investigate changes in both micro- and macronutrient content of the fermented breadfruit beverage pre- and post-fermentation, the cell viability of probiotic strains and native microbiota, additional organic acid measurements, and the survivability of probiotic microorganisms when carrying out further shelf-life trials.

## Figures and Tables

**Figure 1 foods-08-00214-f001:**
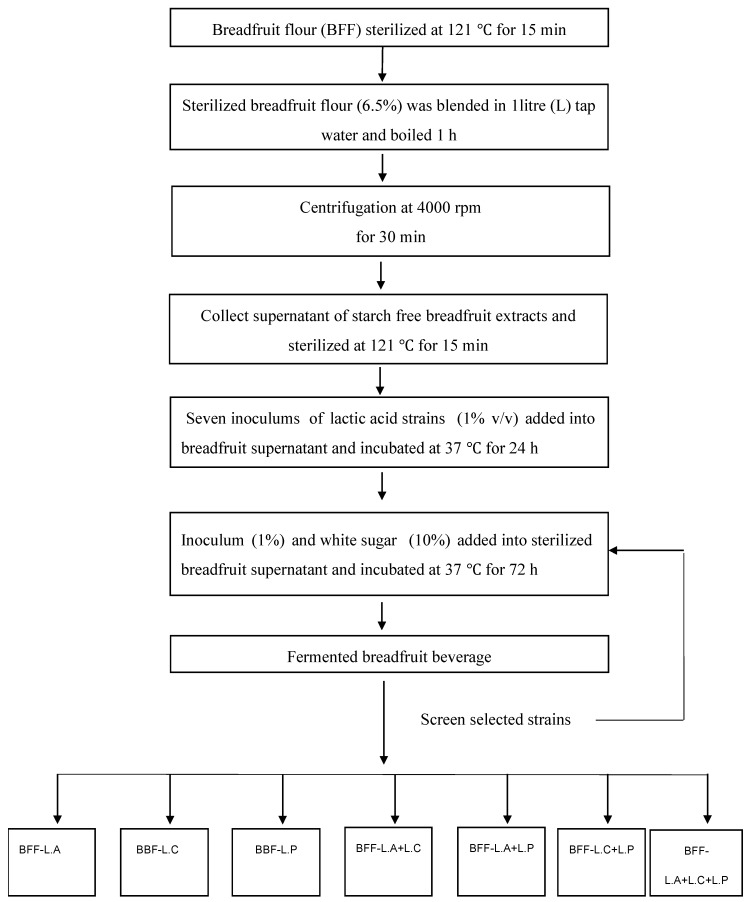
Production of seven *Lactobacilli*-fermented beverage formulations using breadfruit flour as a substrate. L.A (*Lactobacillus acidophilus*), L.C (*Lactobacillus casei*) and L.P (*Lactobacillus plantarum DPC 206*) were both used as monocultures and mixtures (i.e., L.A + L.C + L.P).

**Figure 2 foods-08-00214-f002:**
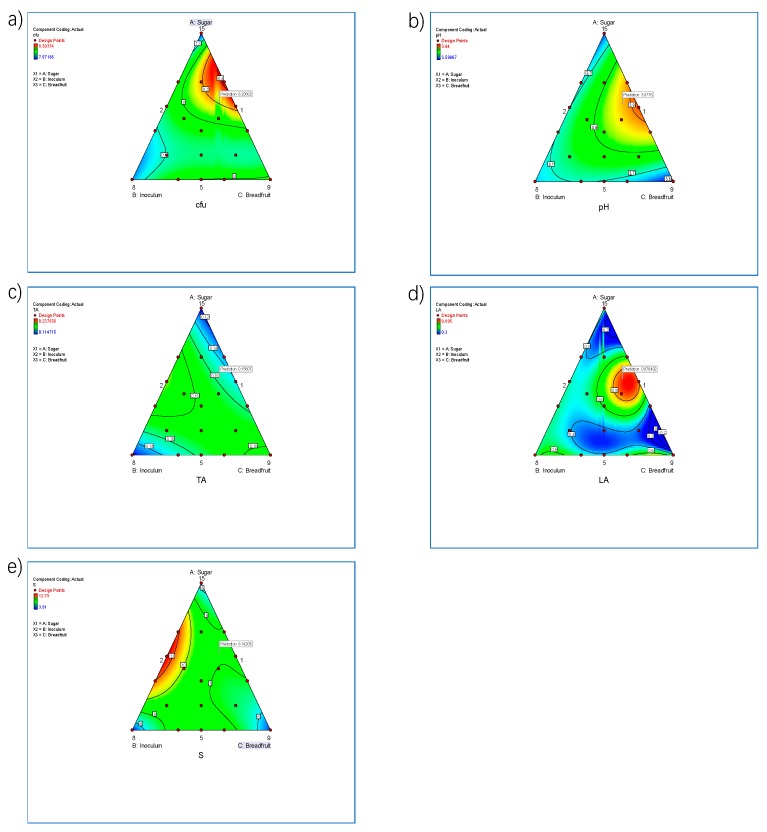
Contour plot showing the effect of sugar, inoculum and breadfruit flour concentration on the CFU (**a**), pH (**b**), titratable acidity (TA) (**c**), lactic acid (LA) (**d**), and sucrose (S) concentration (**e**) in the 48-h-fermented beverage.

**Figure 3 foods-08-00214-f003:**
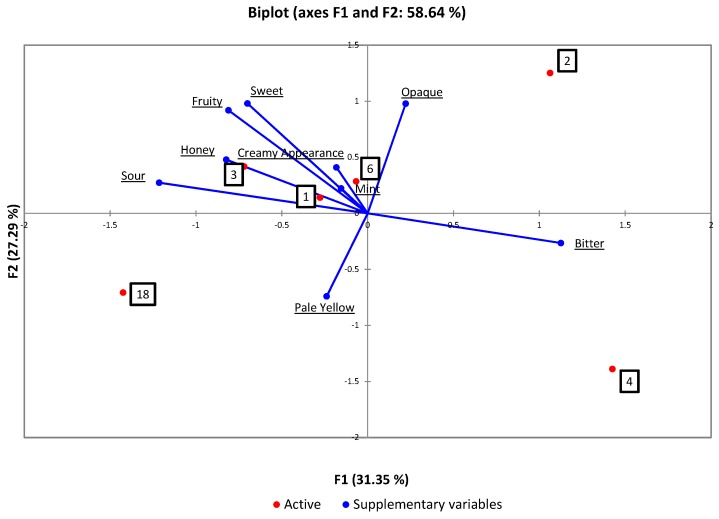
Sample configuration in the first and second dimensions of the Principal Components Analysis plot obtained from projective mapping data. The main sensory attributes were projected as supplementary variables in the analysis. Formulations 1, 2, 3, 4, 6, and 18 were analysed.

**Figure 4 foods-08-00214-f004:**
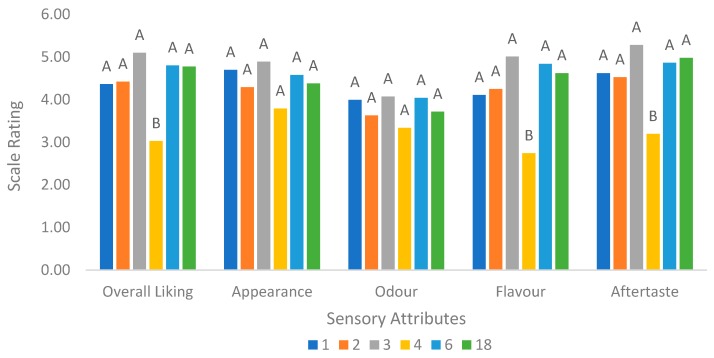
Hedonic testing carried out based on liking of appearance, odour, flavour, aftertaste and overall liking. Values labelled with a different letter represent significant differences (*p* < 0.05) according to the Tukey’s multiple range comparison test. Formulation 1: 5%S, 1%I, 2%BF at 30 °C, Formulation 2: 15%S, 3%I, 2%BF at 30 °C, Formulation 3: 15%S, 1%I, 7%BF at 30 °C, Formulation 4: 5%S, 3%I, 7%BF at 30 °C, Formulation 6: 15%S, 1%I, 7%BF at 37 °C, and Formulation 18: 10%S, 2%I, 4.5%BF at 33.5 °C, where S: Sugar concentration, I: Inoculum concentration, BF: Breadfruit concentration.

**Table 1 foods-08-00214-t001:** Experimental design for formulation of probiotic beverages in this study.

Experiment Number	Sugar Concentration (% *v*/*v*)	Inoculum Concentration (% *v*/*v*)	Breadfruit Concentrations (% *v*/*v*)	Temperature (°C)
1	5	1	2	30
2	15	3	2	30
3	15	1	7	30
4	5	3	7	30
5	15	3	2	37
6	15	1	7	37
7	5	1	2	33.5
8	10	1	2	30
9	5	1	2	30
10	15	3	2	30
11	15	1	7	30
12	5	3	7	30
13	15	3	2	30
14	15	1	7	37
15	5	1	2	33.5
16	10	1	2	30
17	10	2	4.5	33.5
18	10	2	4.5	33.5
19	10	2	4.5	33.5

**Table 2 foods-08-00214-t002:** The number of bacteria cells in fermented breadfruit beverage over 72 h of fermentation.

Bacteria Species (BS)	Fermentation Time (FT)/H (Log_10_ CFU/mL)					F Value		
	0	12	24	48	72	BS	FT	BS × FT
*L. acidophilus*	5.275 ± 0.052 ^Cd^	6.761 ± 0.031 ^Bc^	6.846 ± 0.031 ^Bbc^	7.379 ± 0.338 ^Ab^	8.029 ± 0.096 ^Aa^	5.82 *	96.7 *	1.155
*L. casei*	6.055 ± 0.222 ^Ab^	7.48 ± 0.474 ^Aa^	7.597 ± 0.432 ^Aa^	7.845 ± 0.239 ^Aa^	7.952 ± 0.247 ^Aa^			
*L. plantarum* DPC 206	5.555 ± 0.093 ^BCb^	7.856 ± 0.157 ^Aa^	7.888 ± 0.156 ^Aa^	7.931 ± 0.118 ^Aa^	7.764 ± 0.121 ^Aa^			
*L. acidophilus + L. casei*	5.857 ± 0.091 ^ABb^	7.994 ± 0.188 ^Aa^	8.014 ±0.217 ^Aa^	7.644 ± 0.571 ^Aa^	7.435 ± 0.688 ^Aa^			
*L. acidophilus + L. plantarum* DPC 206	5.955 ± 0.231 ^ABb^	7.826 ± 0.196 ^Aa^	7.872 ± 0.103 ^Aa^	7.962 ± 0.173 ^Aa^	7.701 ± 0.297 ^Aa^			
*L. casei + L. plantarum* DPC 206	5.868 ± 0.031 ^ABb^	7.957 ± 0.091 ^Aa^	8.126 ± 0.122 ^Aa^	8.220 ± 0.166 ^Aa^	7.998 ± 0.229 ^Aa^			
*L. acidophilus+ L. casei + L. plantarum* DPC 206	5.847 ± 0.139 ^ABb^	8.003 ± 0.142 ^Aa^	8.098 ± 0.083 ^Aa^	8.238 ± 0.112 ^Aa^	8.106 ± 0.198 ^Aa^			

The values given above are reported as means and standard deviations. Values with a different letter are significantly different (*p* < 0.05) according to the Fisher’s Least Significant Difference (LSD) post hoc test. Uppercase superscript represent a statistically significant effect within column and lowercase superscripts across each row. * Symbol represents *p* value (**p* < 0.01).

**Table 3 foods-08-00214-t003:** The changes in pH during fermentation in of breadfruit (5%) beverage over 72 h.

Bacteria Species (BS)	Fermentation Time (Hour)					F Value		
	0	12	24	48	72	BS	FT	BS × FT
*L. acidophilus*	5.41 ± 0.02 ^ABa^	5.24 ± 0.01 ^Ab^	4.94 ± 0.02 ^Ac^	4.70 ± 0.05 ^Ad^	4.62 ± 0.09 ^Ad^	92.1 ***	549.8 ***	5.5 ***
*L. casei*	5.38 ± 0.05 ^ABa^	4.3 ± 0.07 ^Bb^	4.06 ± 0.09 ^Bbc^	3.84 ± 0.19 ^Bc^	3.7 ± 0.10 ^BCc^			
*L. plantarum* DPC 206	5.43 ± 0.03 ^Aa^	4.18 ± 0.04 ^BCb^	3.92 ± 0.03 ^Bc^	3.68 ± 0.01 ^Bd^	3.55 ± 0.05 ^BCe^			
*L. acidophilus + L. casei*	5.40 ± 0.00 ^ABa^	4.34 ± 0.11 ^Bb^	4.05 ± 0.19 ^Bbc^	3.84 ± 0.21 ^Bbc^	3.72 ± 0.13 ^Bc^			
*L. acidophilus + L. plantarum* DPC 206	5.39 ± 0.02 ^ABa^	4.13 ± 0.02 ^Cb^	3.95 ± 0.06 ^Bc^	3.69 ± 0.06 ^Bd^	3.57 ± 0.01 ^BCd^			
*L. casei + L. plantarum* DPC 206	5.37 ± 0.02 ^Ba^	4.06 ± 0.01 ^Cb^	3.81 ± 0.05 ^Bc^	3.53 ± 0.04 ^Bd^	3.47 ± 0.00 ^Cd^			
*L. acidophilus+ L. casei + L. plantarum* DPC 206	5.38 ± 0.03 ^ABa^	4.09 ± 0.02 ^Cb^	3.82 ± 0.06 ^Bc^	3.58 ± 0.04 ^Bd^	3.49 ± 0.05 ^BCd^			

The values given above are reported as means and standard deviations. Values with a different letter are significantly different (*p* < 0.05) according to the Fisher’s Least Significant Difference (LSD) post hoc test. Uppercase superscripts represent a statistically significant effect within column and lowercase superscripts across each row. *** Symbol represents *p* value (*** *p* < 0.0001).

**Table 4 foods-08-00214-t004:** Cubic, quadratic and quartic models obtained from the D-optimal design model.

Response	Equation
**CFU**	CFU = 0.075692 × A ^a^ + 0.076848 × B ^a^ + 0.080568 × C ^a^ + 1.01665E-004 × AB +1.75774E-004 × AC + 5.32542E-005 × BC − 5.77027E-006 × ABC + 1.21489E-006 × AB(A−B) + 4.20509E-006 × AC(A−C) ^a^ + 5.41514E-007 × BC(C−B)
**pH**	pH = 0.036188 × A + 0.036682 × B + 0.035848 × C + 2.24981E-005 × AB +1.25414E-004 × AC ^a^ + 2.76765E-005 × BC
**TA**	TA = 1.12385E-003 × A + 1.16534E-003 × B + 1.85367E-003 × C + 3.27095E-005 × AB ^a^ − 2.09864E-006 × AC + 5.00860E-006 × BC
**LA**	LA = 0.53 × A + 0.53 × B + 0.48 × C − 0.13 × AB +1.23 × AC − 0.30 × BC − 0.55 × AB(A−B) + 1.78 × AC(A−C) − 0.57 × BC(B−C) + 13.05 × A^2^BC − 18.68 × AB^2^C + 6.39 × ABC^2^ − 0.73 × AB(A−B)^2^ − 12.09 × AC(A−C)^2 a^ + 3.39 × BC(B−C)^2^
**S**	S = 0.050304 × A + 0.043473 × B + 0.042798 × C + 3.32747E-003 × AB ^a^ +1.51232E-003 × AC + 2.02706E-003 × BC – 1.02840E-004 × ABC

A = sugar, B = inoculum, C = breadfruit. Lowercase superscript ^a^ represents a statistically significant effect (*p* < 0.05).

**Table 5 foods-08-00214-t005:** ANOVA of the regression models and regression coefficients for parameter used in the optimisation of fermented breadfruit beverages. A = sugar, B = inoculum, C = breadfruit. ** *p* < 0.01; * 0.01 ≤ *p* < 0.05; *p* ≥ 0.10.

Response	Model	A	B	C	AB	AC	BC	ABC	AB(A−B)	AC(A−C)	BC(B−C)	A^2^BC	AB^2^C	ABC^2^	AB(A−B)^2^	AC(A−C)^2^	BC(B−C)^2^	Mean Experimental Value	Predicted Model Value
**CFU**	Cubic	7.57 *	7.68 *	8.06 *	1.02	1.76 *	0.53	−5.77	1.21	4.21 **	0.54	---	---	---	---	---	---	7.924 log CFU/mL	8.208 log CFU/mL
**pH**	Quadratic	3.62	3.67	3.58	0.22	1.25 **	0.28	---	---	---	---	---	---	---	---	---	---	3.82	3.877
**TA**	Quadratic	0.11	0.12	0.19	0.33 **	−0.02	0.05	---	---	---	---	---	---	---	---	---	---	0.177%	0.156%
**LA**	Quartic	0.53	0.53	0.48	−0.13	1.23	−0.30	---	−0.55	1.78	−0.57	13.05	−18.68	6.39	−0.73	−12.09 *	3.39	0.70 g/mL	0.87 g/mL
**S**	Special Cubic	5.03	4.35	4.28	33.27 **	15.12	20.27	−102.84	---	---	---	---	---	---	---	---	---	8.373%	8.142%

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
