# Peer review of "Development of a Probiotic Beverage Using Breadfruit Flour as a Substrate"

_foods, 2019, doi:10.3390/foods8060214_

Round 1
Reviewer 1 Report
The study is well designed and presented. Moreover, the paper has a scientific novelty in terms fermentation of due to breadfruit use. It was been a pleasure reading. I advise only some minor corrections.
In M&M section LAB in MRS are usually cultivated by agar pour plate method instead of spread.
In lines 180-192 pls correct all strain in italic form
In table 5 first line check for 'cut' word.
In figure 2 non of the presented data are readable in the presented format.
Author Response
Reviewer 1
The study is well designed and presented. Moreover, the paper has a scientific novelty in terms fermentation of due to breadfruit use. It was been a pleasure reading. I advise only some minor corrections.
In M&M section LAB in MRS are usually cultivated by agar pour plate method instead of spread.
This has been amended, the word ‘spread’ has been changed to ‘pour’.
In lines 180-192 pls correct all strain in italic form
This has been corrected.
In table 5 first line check for 'cut' word.
This was due to the typesetting done by the journal, we apologise. It has now been corrected.
In figure 2 non of the presented data are readable in the presented format.
A higher resolution metafile image has been submitted to the journal in word file format – but may not be so high resolution in PDF form, perhaps the assistant editor can provide a higher resolution file to the reviewer.
Reviewer 2 Report
General remarks.
The title suggest usage probiotic strain in the experiment but there is no information about this in the text. Not all LAB are probiotics.
The aim of the study should refer to breadfruit flour.
There are a few stylistic and editorial errors, i.e. not all LAB names are written italic.
Section 2.1
The sentence in the lines 61-62 should be rewritten to emphasize flour production.
Section 2.2
The term "types" is imprecise. The autors named one starin (Lactobacillus plantarum DPC206) and two genera (Lactobacillus acidophilus and Lactobacillus casei). The probiotic features are associated with strains, because of that the authors should add information about which strain of L. acidophilus and L. casei was used. And, of course, which probiotic feature/features they possess.
In the line 67, the manufacturer's name of the MRS broth should be given.
Section 2.3.1
There is no detailed information about stock cultures: the conditions in which they were propagated, whether they were centrifuged and how.
The production diagram should contain information that monoculture and mixture were used.
In the line 84, the manufacturer's name of the MRS agar should be given.
Line 141: should be: products were also described
Line 197: should be: culture or their mixtures
Author Response
Reviewer 2
General remarks.
The title suggest usage probiotic strain in the experiment but there is no information about this in the text. Not all LAB are probiotics.
We agree with the reviewer and have added one statement in the Introduction that reads:
“Strains of lactic acid bacteria (LAB) belonging to the genera Lactobacillus and Bifidobacterium are commonly used as probiotics.”
The aim of the study should refer to breadfruit flour.
The word flour has been added to the sentence, this now reads:
“..this study aims to investigate lactic acid fermentation using breadfruit flour as a substrate and monitor the changes in physicochemical properties..”
There are a few stylistic and editorial errors, i.e. not all LAB names are written italic.
This has been noted by Reviewer 1 and has been amended.
Section 2.1
The sentence in the lines 61-62 should be rewritten to emphasize flour production.
This has been amended and some details has been added.
Lines 63-67: Breadfruit flour was sourced from a local company (Maiden South Pacific Company, New Zealand) located in Auckland, New Zealand. The breadfruit flour was imported from Natural Foods International Limited, Samoa. Mature breadfruit harvested were peeled, cored and sliced before being air dried, and then hand milled into powder form (G. Percival, personal communication, January 25, 2019).
Section 2.2
The term "types" is imprecise. The authors named one strain (Lactobacillus plantarum DPC206) and two genera (Lactobacillus acidophilus and Lactobacillus casei). The probiotic features are associated with strains, because of that the authors should add information about which strain of L. acidophilus and L. casei was used. And, of course, which probiotic feature/features they possess.
The word type has been deleted as it is imprecise. Additional information on the strains has been added.
In the line 67, the manufacturer's name of the MRS broth should be given.
This has been added accordingly,“..MRS broth (Merck, Darmstadt, Germany)”
Section 2.3.1
There is no detailed information about stock cultures: the conditions in which they were propagated, whether they were centrifuged and how.
The production diagram should contain information that monoculture and mixture were used.
This information is now added in the figure caption. The sentence reads:
“L.A (Lactobacillus acidophilus), L.C (Lactobacillus casei) and L.P (Lactobacillus plantarum DPC206) were both used as monocultures and mixtures (i.e. L.A+L.C+L.P).”
In the line 84, the manufacturer's name of the MRS agar should be given.
This has been added accordingly.
Line 141: should be: products were also described
This has been changed accordingly.
Line 197: should be: culture or their mixtures
This has been changed accordingly.
Reviewer 3 Report
The paper is original and the topic is interesting.
However, this manuscript needs some improvements before it could be accepted for publication.
Abstract
Line 17: “formulation 4”. Explain.
Material and methods
2.8 and 2.9 (Lines 122-137): These determinations are not well explained. Sugars are only sucrose? The determination of lactic acid is very difficult to understand. Are there other compounds detected? In order to determine lactic acid it is not necessary GC-MS.
Results and Discussion
Table 5. Respons-e, Quadrati-c. These words are separated. The table design should be improved.
Author contributions?
References.
They should be described according journal rules.
Author Response
Reviewer 3
The paper is original and the topic is interesting.
However, this manuscript needs some improvements before it could be accepted for publication.
Abstract
Line 17: “formulation 4”. Explain.
Details on formulation 4 has been added. This formulation contained 5% sugar, 3% inoculum, 7% breadfruit flour at 30oC.
Material and methods
2.8 and 2.9 (Lines 122-137): These determinations are not well explained. Sugars are only sucrose? The determination of lactic acid is very difficult to understand. Are there other compounds detected? In order to determine lactic acid it is not necessary GC-MS.
The following has been added to the manuscript to clarify the methods for sugar:
Lines 129-131: This method was capable of analysing for the presence of fructose, glucose, lactose, maltose and sucrose sugars. However, sucrose was the only sugar detected in the samples and thus the only sugar reported in this study.
We agree with the reviewer that it is not necessary to carry out determination of lactic acid a GCMS method. We have actually used the method of titratable acidity described in Section 2.7 Line 119 for determination of lactic acid in this study. Titratable acidity does not target a specific acid, but as lactic acid fermentation was carried out, the samples in this study contained mainly lactic acid. Although there may be other acids, these other acids are only present in minute quantities. For our purposes (and convention), we have assumed that 100% lactic acid in the sample for the titration carried out in our study.
Results and Discussion
Table 5. Respons-e, Quadrati-c. These words are separated. The table design should be improved.
This has been changed accordingly.
Author contributions?
This information has been added.
References.
They should be described according journal rules.
The journal has subtyped the references and should be according to the journal’s standard now.
Round 2
Reviewer 3 Report
The changes recommended by the reviewers have been done.